# Neurally boosted supervised spectral clustering

## Abstract

Network embedding methods compute geometric representations of graphs that render various prediction problems amenable to machine learning techniques. Spectral network embeddings are based on eigenvectors of a normalized graph Laplacian. When coupled with standard classifiers, spectral embeddings yield good baseline performance in node classification tasks, but they are widely considered to have been amply surpassed by newer embedding methods and Graph Convolutional Networks. In this work, we provide a theory-informed implementation of supervised spectral clustering and maximize its performance with the use of a carefully designed RADIAL classifier that leverages fundamental geometric properties of spectral embeddings. RADIAL raises the performance of supervised spectrally clustering to previously unobserved levels and renders it directly competitive to newer methods in the context of transductive node classification on social networks. We also observe that that features learned in an inner layer of RADIAL can significantly enhance the performance of standard Graph Convolutional Networks.

## 1 Introduction

Spectral clustering is a classical unsupervised algorithm for clustering in graphs. It gained prominence in data mining and machine learning approximately two decades ago (Shi & Malik, 2000; Ng et al., 2001) and it has since found many applications in multiple domains. While originally conceived as an unsupervised method, spectral clustering has also been employed in transductive learning, and in particular in semi-supervised node classification that is the focus of this work.

**Background.** The work of Tang & Liu (2011) was likely what initiated a long line of work on multi-label classification with experimental focus on social network benchmarks. They described a general framework for node classification consisting of two steps: (i) computing a network embedding, (ii) using the embedding as input to a standard classifier. They instantiated step (i) with a spectral embedding, giving thus rise to what can be viewed as a *supervised spectral clustering* algorithm. Albeit simple, the algorithm proved to be superior relative to the then-available baselines. The thread continued with the very prominent DeepWalk embedding algorithm (Perozzi et al., 2014), that claimed superiority over spectral clustering which however remained competitive. The success of DeepWalk unleashed a wave of research works on network embeddings, including (Grover & Leskovec, 2016; Tang et al., 2015; Tsitsulin et al., 2018; Qiu et al., 2018a), and a race for higher classification accuracy [1]. More recently, network embedding methods were succeeded by graph neural networks (GNNs) (e.g. see (Kipf & Welling, 2017; Velickovic et al., 2018)) that have dominated the competition. Since the publication of DeepWalk, supervised spectral clustering was at-best considered a baseline, or entirely vanished from consideration. However it has recently made a comeback with the work of Huang et al. (Huang et al., 2021) that described a simple 'correction and smooth' (C&S) procedure that can enhance the output of other models. Some of the state-of-the-art results in in the Open

---

[1] A very detailed list can be found in `https://github.com/chihming/awesome-network-embedding`

Graph Benchmark leaderboard (Hu et al., 2021) are now based on variants of supervised spectral clustering in combination with the C&S process. These easily surpass previous embedding methods and even exceed methods based on GNNs using only a small fraction of their number of trainable parameters.

**A Critical Look.** This work originates from the observation that previous literature has treated spectral clustering somewhat casually, and largely ignoring recent major advances in its theoretical understanding. As discussed above, supervised spectral clustering consists of two steps: (i) Computing a spectral embedding, and (ii) Giving the embedding as input to a multi-label classifier. In the literature, the term 'spectral clustering' is used for multiple different algorithms that implement steps (i) and (ii) in various and sometimes not adequately described ways.

For example, for step (i), (Tang & Liu, 2011) and (Perozzi et al., 2014) use eigenvectors of the normalized graph Laplacian as proposed in (Ng et al., 2001) but appear to omit a normalization step from that algorithm, (Huang et al., 2021) also use the normalized graph Laplacian but with an additional 'rotation' transformation, while other works use the random walk matrix (i.e. the non-symmetric normalization of the Laplacian) which in fact is a more accurate relaxation of the underlying NCut problem (Shi & Malik, 2000). Classifiers used for step (ii) include an One-vs-All SVM-based model (Tang & Liu, 2011), an One-vs-Rest logistic regression model (Perozzi et al., 2014), a softmax regression model, and a three-layered neural network (Huang et al., 2021). Adding to the confusion, the computation of eigenvectors has been viewed both as computationally expensive to the point of preventing its application on larger networks (Perozzi et al., 2014), or as able to scale up to very large networks via standard iterative eigensolvers whose fast convergence cannot however be guaranteed (Huang et al., 2021).

**Goal.** These considerations provide the motivation of our work. Concretely, we seek to provide a theory-informed implementation of supervised spectral clustering, maximize its performance with the use of a carefully designed neural model and study its practical performance in the context of multi-label classification on social networks.

**Contributions.** Our RADIAL classifier can be conceptually viewed as a 'neural' implementation of the seminal theoretical work of Lee et al. (2014b) on multi-way Cheeger inequalities for spectral clustering. These theoretical guarantees are based on a variant of spectral clustering that is not well-known or widely used, yet it can be theoretically analyzed without assumptions about the underlying graphs. The analysis of Lee et al. applies to *unsupervised* clustering. Thus a key contribution of our work is to transfer the geometric intuition developed in (Lee et al., 2014b) to the *supervised* setting with the design of custom-made neural classifiers that respect and exploit the geometric facts.

Indeed, RADIAL raises the performance of supervised spectral clustering on transductive node classification to previously unobserved levels, sometimes more than 15% higher than vanilla classifiers. As our ablation study confirms, this performance is achieved by introducing certain modifications and enhancements to the basic unsupervised algorithm of (Lee et al., 2014b), and other training tricks.

Within our experimental context, when only *graphic data* are taken as input, we show that when training nodes are scarce, supervised spectral clustering is directly competitive to prominent alternative methods that were thought to be vastly superior. On the other hand, we observe that non-parametric diffusion-based methods become progressively more effective when training points are abundant, thus offering an empirical explanation of the success of the C&S algorithm (Huang et al., 2021). Moreover, based on intuition derived in section 3.4, we develop a method of enhancing the performance of standard GCNs (Kipf & Welling, 2017). More specifically, in section 4, we show that when *node features* are available, the performance of GCNs can be significantly enhanced by feeding as additional input features learned by our RADIAL classifier.

## 2 SPECTRAL CLUSTERING WITH SUPERVISION: THE ALGORITHM

### 2.1 SPECTRAL EMBEDDING AND GEOMETRIC INTUITION

We begin with a review of the spectral embedding algorithm. Along it we give concrete details on how it can be implemented to run in near-linear time with strong convergence guarantees. In Section 2.1.3 we discuss the geometric intuition that underlies the theoretical analysis in (Lee et al., 2014b). We note that the material in this subsection is not new. We present it for completeness and in particular in order to discuss the geometric intuition which plays a key role in the construction of our classifiers.

#### 2.1.1 DEFINITIONS AND TERMINOLOGY

Let $G = (V, E, w)$ be a weighted undirected graph.

**Definition 1** (**Cut and Volume**). *If $S \subset V$, we define the cut of $S$ to be*

$$cut(S) = \sum_{i \in S, j \notin S} w_{i,j}$$

*We also define the volume of $S$ to be*

$$vol(S) = \sum_{s \in S} cut(v).$$

*We will often use the term **degree** for the cut of a single node.*

**Definition 2** (**Conductance**). *The conductance of a cluster $S \subset V$ is defined by*

$$\phi(S) = \frac{cap(S, V - S)}{\min(vol(S), vol(V - S))}$$

We denote by $D$ the diagonal matrix of the degrees, by $d$ the vector of the node degrees, and by $L = D - A$ the graph Laplacian, where $A$ is the adjacency matrix of the graph.

#### 2.1.2 SPECTRAL EMBEDDING AND THE UNSUPERVISED ALGORITHM

The combinatorial idea that underlies spectral clustering is that good clusters should have a relatively small volume to their exterior, or more specifically a low conductance. More concretely, the goal is to find $k$ disjoint clusters $\{S_1, S_2, ...S_k\}$ such that $\max_i \phi(S_i)$ is as small as possible. In this paper we will be looking for $k$ clusters that also provide a partitioning of the vertex set, i.e. we have $\bigcup_i S_i = V$. The partitioning problem is less well-understood (Louis et al.), but its simpler non-partitioning variant has been studied extensively from a theoretical perspective (Lee et al., 2014b), and approximation guarantees are known for it. The spectral clustering algorithm is broadly understood as a relaxation of the discrete clustering problem (e.g. see (Cucuringu et al., 2016) for a complete discussion), and a number of variants have been proposed (von Luxburg, 2007). In its simplest form, the variant that is analyzed in (Lee et al., 2014b) consists of three steps, summarized here:

*(a) Spectral Embedding.* Compute the eigenvectors $x_j$ corresponding to the $k$ smallest non-zero eigenvalues of the generalized problem $Lx = \lambda Dx$, under the constraint that $x^T d = 0$. The eigenvectors are normalized so that $x_j^T Dx_j = 1$. Let $X \in \mathbb{R}^{n \times k}$ be the matrix whose columns are the $k$ eigenvectors[2].

*(b) Orthogonal Projection.* Let $X' = XR$, where $R' \in \mathbb{R}^{k \times k'}$ a random orthogonal projection matrix, where $k' \leq k$.

---

[2]Spectral embeddings can be computed in nearly-linear time. We discuss their computation in Appendix C.

*(c) Radial Projection.* Let $Y$ be the matrix resulting by dividing each row of $X$ by its Euclidean norm, i.e. by 'radially projecting' each point onto the unit sphere in $\mathbb{R}^k$. Each row of $Y$ becomes the embedding of the corresponding node.

*(d) Geometric Partitioning.* Use some unsupervised geometric clustering algorithm on the embedding $Y$. (e.g. k-means, or the provable algorithms presented in (Lee et al., 2014b)).

We remark that the two hyperparameters $k$ and $k'$ are further specified in sections 2.2 and 3.1.

### 2.1.3 THE GEOMETRIC INSIGHT

The radial projection step reflects the geometric intuition that the eigenvectors place the nodes into $k$ *directions* in the embedding space.

Consider the eigenvector matrix $X \in \mathbb{R}^{n \times k}$. These generalized eigenvectors are $D$-orthogonal (Stewart & Sun, 1990), i.e. we have $x_i^T D x_j = 0$, for each $i \neq j$. Given that the eigenvectors are also normalized, we have $X^T D X = I$.

Using a well known identity we have

$$trace(X^T D X) = trace(D X X^T) = k.$$

Summing up the $n$ diagonal entries of $D X X^T$ we thus get

$$\sum_{j=1}^{n} d_j ||x_j||_2^2 = k. \tag{1}$$

Now pick an arbitrary direction in $\mathbb{R}^k$, i.e. a unit vector $z$. Let $w = Xz$. Since $X^T D X = I$, and $z^T z = 1$ we have

$$z^T X^T D X z = w^T D w = \sum_{j=1}^{n} d_j w_j^2 = 1. \tag{2}$$

Let us now take a closer look at the above equations. From equation 1 we get that that the $d$-weighted mass of the $n$ embedding points is $k$. On the other hand, note that $w_j^2$ is the norm of the projection of $X[j]$ onto $z$. Thus, equation 2 shows that the $d$-weighted mass of the projections of the $n$ points onto $z$ is always equal to 1, for any direction $z$.

A further interpretation of the above equations is that any given direction $z$ captures only a $1/k$ fraction of the $d$-weighted sum of norms of the $n$ embedding points. Hence the $d$-mass of the points must concentrate in at least $k$ different directions. While finding such directions is not straightforward, (Lee et al., 2014b) proves that radially projecting the points onto the unit sphere has the following property: if $S$ is a set of points in $Y$ that are in a ball of a small diameter in $\mathbb{R}^k$, then $S$ also cannot concentrate a large fraction of the $d$-mass of the points, and thus the points in $Y$ are expected to be in $k$ well-separated spheres in $\mathbb{R}^k$. This specific geometric arrangement facilitates the design of an unsupervised geometric clustering algorithm on the points of $Y$ in (Lee et al., 2014b), but it will also guide the design of our supervised geometric clustering model.

## 2.2 NEURAL CLASSIFIERS

We present three types of neural classifiers that reflect our discussion in section 2.1. These are different but they share certain layers and so they are presented in a compact way in Figure 1. Below we make our main points on the design of these classifiers.

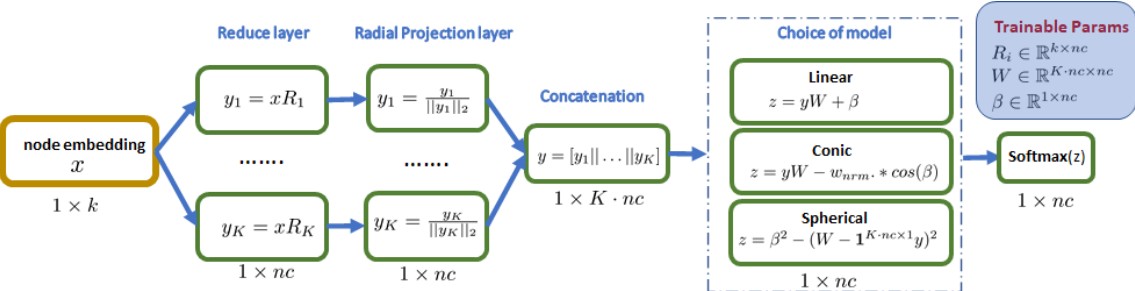

Figure 1: The forward function of the RADIAL classifiers. Pseudocode is given in Appendix section D. Details of the algorithm are discussed in section 2.2. Section 2.3 considers initialization and training tricks. We note that $k$ is the embedding dimension, $K$ is the number of channels, and $nc$ is the number of classes.

Step (ii) in the unsupervised algorithm described in section 2.1.2 is a random orthogonal projection. In analogy, we introduce a 'Reduce Layer' which in our case is entirely **trainable**. The Reduce and Radial Projection layers are reproduced independently in $K$ independent channels, where $K$ is a hyperparameter. Each of these channels has its own operator $R_i$.

We propose three different types of classifiers: *Conic*, *Linear*, and *Spherical*. These share the same parameters $W$ and $\beta$. The outer dimension of $W$ is proportional to the number of channels, but the inner dimension of both $W$ and $\beta$ are equal to the number of classes. What differentiates these three types of classifiers is the role of $W$ and $\beta$. Their names reflect the corresponding geometric shape of the decision boundaries.

*Linear:* Each of the $nc$ rows of $W$ defines a hyperplane in $\mathbb{R}^{K \times nc}$ as its normal vector. The decision boundary for row $W_j$ is the hyperplane $yW + \beta$.

*Conic:* Row $W_j$ represents represents a direction in $\mathbb{R}^{K \times nc}$. The decision boundary is given by equation $yW_j - ||W_j||_2 cos(\beta)$, which is well-understood to describe the surface of a cone with axis $W_j$ and aperture $\beta$, given that $y$ has unit norm (line 12). We note that the decision boundary can be rewritten as $yW_j + \beta'$, where $\beta' = ||W_j||_2 cos(\beta)$. Thus it can be seen that *Conic* is a restricted or 'regularized' version of *Conic* because it enforces $\beta' \leq ||W_j||_2$.

*Spherical:* Row $W_j$ represents a point in $\mathbb{R}^{K \times nc}$, which is the center of the sphere with of radius $\beta$. The decision boundary is the sphere $\beta^2 - ||W_j - y||_2^2$.

## 2.3   INITIALIZATION AND TRAINING

The discussion in previous sections allows us certain considerations on the initialization of the trainable parameters. In the unsupervised algorithm, the Reduce Layer is taken to be a random orthogonal projection, and we can thus initialize it as such. Moreover, according to the discussion in section 2.1.3, the node embeddings given by $XR$ are expected to concentrate in at least $nc$ different directions/geometric clusters after the radial projection step. Approximate directions or 'centers' for these clusters can be found using the spectral embeddings of the training points. Specifically, in order to compute such approximate centers or directions, we take the radial projection of the training points for each cluster separately, compute their average, and then we radially project it. These initializations can be carried out independently for each of the $K$ channels. In our experiments we have used such initializations; they do not affect accuracy, but seem to improve convergence speed (see Appendix C.1).

We also explore a pre-training strategy for $K$-channel models. We first train $K$ single-channel models with different random initializations. The Reduce layers from these $K$ models are used to initialize the corresponding layers of the $K$-channel model, while the last layer (parameter matrix $W \in \mathbb{R}^{K \cdot nc \times nc}$) is initialized randomly.

## 3 EXPERIMENTS

This section summarizes our experiments. Our comments on the implications of these experiments are given as bullet-points.

### 3.1 EXPERIMENTAL SETUP

**Datasets, Hyperparameters, Variance.** We use well-known benchmarks and follow standard practices. We refer the reader to Appendix A for detailed information.

**Embedding Dimension.** We uniformly set the embedding dimension (parameter $k$ in Figure 1) to $k = 2 \cdot nc$. We note that the performance of all methods can be slightly raised with a choice of higher dimension, but we stick with a choice proportional to the number of classes in order to conform with the underlying theory.

**Evaluation Method.** We measure the accuracy of the various methods trained with randomly selected training sets at label rates: $2^j \%$, for $j = 0, \ldots, 5$. For each label rate, we pick 5 different random training sets. For each of these training sets we train each classifier for 5 different random initializations of the trainable weights. For each such experiment, we perform 3-fold cross validation where each validation set is selected randomly and consists of 20% of the training set (i.e. 20% of the label rate). We ensure that every class is represented in our training set by forcing one random node from each class in the training set. Validation is used for RADIAL in order to select the best-performing model over the course of training.

### 3.2 ABLATION STUDY

In Figure 2, we present an ablation study on the layers of the RADIAL classifier.

| | CLASSIFIER | \multicolumn{6}{c}{CITESEER} | | | | | | \multicolumn{6}{c}{CORA} | | | | | | \multicolumn{6}{c}{WIKICS} | | | | | |
|---|---|---|---|---|---|---|---|---|---|---|---|---|---|---|---|---|---|---|---|
| | | 1% | 2% | 4% | 8% | 16% | 32% | 1% | 2% | 4% | 8% | 16% | 32% | 1% | 2% | 4% | 8% | 16% | 32% |
| 1 | NO-RADIAL | 56.4 | 50.3 | 51.0 | 54.3 | 57.4 | 58.4 | 16.7 | 23.9 | 25.7 | 28.6 | 26.8 | 29.0 | 29.7 | 34.3 | 35.0 | 37.7 | 35.8 | 35.9 |
| 2 | NO-REDUCE | 49.0 | 63.6 | 66.9 | 68.9 | 68.9 | 71.1 | 18.9 | 18.8 | 18.8 | 17.5 | 25.5 | 27.4 | 69.4 | 71.7 | 73.9 | 74.8 | 75.8 | 76.2 |
| 3 | CONIC | 45.8 | 59.6 | 64.6 | 67.6 | 69.7 | 70.7 | 16.0 | 19.0 | 19.0 | 22.7 | 24.4 | 26.7 | 69.9 | 73.2 | 75.0 | 75.4 | 78.2 | 78.7 |
| 4 | SPHERICAL | 45.0 | 62.3 | 65.1 | 67.9 | 69.0 | 70.9 | 18.1 | 19.6 | 22.3 | 24.5 | 26.3 | 28.1 | 68.8 | 73.2 | 75.3 | 76.9 | 78.5 | 79.0 |
| 5 | CONIC-2C | 48.1 | 60.9 | 65.3 | 68.2 | 69.9 | 71.1 | 17.1 | 17.9 | 20.2 | 21.7 | 21.7 | 27.0 | 70.5 | 74.0 | 75.0 | 76.1 | 78.4 | 79.8 |
| 6 | CONIC-2C-PT | 53.5 | 57.4 | 65.5 | 69.1 | 70.8 | 71.0 | 17.4 | 20.1 | 18.4 | 20.8 | 22.0 | 25.8 | 71.8 | 73.9 | 75.3 | 77.6 | 78.0 | 80.1 |
| | CLASSIFIER | \multicolumn{6}{c}{PUBMED} | | | | | | \multicolumn{6}{c}{ARXIV} | | | | | | \multicolumn{6}{c}{PRODUCTS} | | | | | |
| | | 1% | 2% | 4% | 8% | 16% | 32% | 1% | 2% | 4% | 8% | 16% | 32% | 1% | 2% | 4% | 8% | 16% | 32% |
| 1 | NO-RADIAL | 53.2 | 53.6 | 59.0 | 55.2 | 58.8 | 58.5 | 21.2 | 26.5 | 26.0 | 24.6 | 25.8 | 25.1 | 27.9 | 28.7 | 31.1 | 28.9 | | |
| 2 | NO-REDUCE | 63.5 | 64.6 | 63.4 | 64.3 | 65.2 | 64.0 | 47.6 | 48.5 | 49.0 | 49.3 | 49.3 | 49.4 | 66.6 | 66.7 | 66.6 | 66.7 | | |
| 3 | CONIC | 75.6 | 75.4 | 76.0 | 76.6 | 76.6 | 76.6 | 58.4 | 60.5 | 61.3 | 62.1 | 62.4 | 62.5 | 80.9 | 80.9 | 80.9 | 81.0 | | |
| 4 | SPHERICAL | 75.0 | 75.0 | 75.5 | 76.0 | 76.1 | 76.3 | 57.8 | 60.5 | 61.9 | 63.0 | 63.7 | 63.8 | 82.1 | 82.3 | 82.4 | 82.5 | | |
| 5 | CONIC-2C | 75.2 | 75.3 | 76.0 | 76.4 | 76.8 | 76.8 | 58.6 | 60.7 | 62.6 | 64.1 | 64.9 | 65.4 | 83.5 | 83.8 | 83.9 | 84.0 | | |
| 6 | CONIC-2C-PT | 75.3 | 75.9 | 76.1 | 76.5 | 77.4 | 77.2 | 59.5 | 61.2 | 62.9 | 64.4 | 65.2 | 65.7 | | | | | | |

Figure 2: An ablation study. CONIC-2C includes all layers of RADIAL (fig, 1). The last layer is 'Conic' and $K = 2$. CONIC-2C-PT adds a pre-training strategy (see section 2.3). Details are discussed in section 3.2 and a shorter ablation study on multiple channels can be found in Appendix E.1.

Classifier NO-RADIAL in the Linear branch of RADIAL, *excluding* the Reduce and Radial Projection layers. Classifier NO-REDUCE runs the Linear branch excluding the Reduce layer. The results are similar when Conic or Spherical are used instead. Classifiers CONIC and SPHERICAL include all layers of the RADIAL algorithm in Figure 1.

• We note that despite its very different and non-standard mathematical form, the performance of SPHERICAL is comparable to CONIC, and potentially better. However we have found that it requires more training, and the reported results use 4K epochs of training for SPHERICAL, and 2K epochs of training for SPHERICAL. For that reason, we stick with CONIC for the multi-channel experiments.

• Overall, Figure 2 justifies the layers of RADIAL, especially for larger datasets, and demonstrates the extra power of multiple channels and appropriate pre-training strategies. The CORA dataset is a clear exception that we discuss separately in section 3.4. Further ablation studies are reported in section E.1.

## 3.3 RADIAL VS STANDARD CLASSIFIERS

In Figure 3 we present a comparison of CONIC-2C and CONIC-2C-PT with standard classifiers. As input to these other classifiers, we use the *radially projected* spectral embedding, as it maximizes their performance.

| | CLASSIFIER | CITESEER | | | | | | CORA | | | | | | WIKICS | | | | | |
|---|---|---|---|---|---|---|---|---|---|---|---|---|---|---|---|---|---|---|---|
| | | 1% | 2% | 4% | 8% | 16% | 32% | 1% | 2% | 4% | 8% | 16% | 32% | 1% | 2% | 4% | 8% | 16% | 32% |
| 1 | LIBLINEAR-R | 55.3 | 64.4 | 67.8 | 68.7 | 69.8 | 70.9 | 63.6 | 68.8 | 71.5 | 74.1 | 75.0 | 76.0 | 72.8 | 73.6 | 74.9 | 75.4 | 76.1 | 76.3 |
| 2 | LIBLINEAR-NR | 54.5 | 56.7 | 65.0 | 68.6 | 70.4 | 71.1 | 64.4 | 65.1 | 71.0 | 75.7 | 78.2 | 78.3 | 67.5 | 71.0 | 75.0 | 76.2 | 76.9 | 77.0 |
| 3 | KNN (k=11) | 49.6 | 62.2 | 67.3 | 68.7 | 69.8 | 70.7 | 50.9 | 66.9 | 70.7 | 74.8 | 77.3 | 78.9 | 70.3 | 72.7 | 74.6 | 76.6 | 78.5 | 79.5 |
| 4 | CONIC-2C | 48.1 | 60.9 | 65.3 | 68.2 | 69.9 | 71.1 | 17.1 | 17.9 | 20.2 | 21.7 | 21.7 | 27.0 | 70.5 | 74.0 | 75.0 | 76.1 | 78.4 | 79.8 |
| 5 | CONIC-2C-PT | 53.5 | 57.4 | 65.5 | 69.1 | 70.8 | 71.0 | 17.4 | 20.1 | 18.4 | 20.8 | 22.0 | 25.8 | 71.8 | 73.9 | 75.3 | 77.6 | 78.0 | 80.1 |
| | CLASSIFIER | PUBMED | | | | | | ARXIV | | | | | | PRODUCTS | | | | | |
| | | 1% | 2% | 4% | 8% | 16% | 32% | 1% | 2% | 4% | 8% | 16% | 32% | 1% | 2% | 4% | 8% | 16% | 32% |
| 1 | LIBLINEAR-R | 60.3 | 60.6 | 61.7 | 64.3 | 67.1 | 68.8 | 43.8 | 45.4 | 47.3 | 48.9 | 50.2 | 51.4 | 67.1 | 69.4 | 73.4 | 75.4 | 76.5 | 77.3 |
| 2 | LIBLINEAR-NR | 73.3 | 73.4 | 72.7 | 72.1 | 71.6 | 71.5 | 56.9 | 58.7 | 59.2 | 59.1 | 59.2 | 59.2 | 79.9 | 79.9 | 80 | 80.1 | 80.1 | 80.1 |
| 3 | KNN (k=11) | 73.5 | 74.7 | 75.4 | 76.3 | 77.3 | 78.3 | 54.7 | 57.3 | 59.2 | 60.9 | 62.6 | 64.3 | 80.7 | 81.9 | 83 | 84.1 | 85 | 85.9 |
| 4 | CONIC-2C | 75.2 | 75.3 | 76 | 76.4 | 76.8 | 76.8 | 58.6 | 60.7 | 62.6 | 64.1 | 64.9 | 65.4 | 83.5 | 83.8 | 83.9 | 84 | | |
| 5 | CONIC-2C-PT | 73 | 75.9 | 76.1 | 76.5 | 77.4 | 77.2 | 59.5 | 61.2 | 62.9 | 64.4 | 65.2 | 65.7 | | | | | | |

Figure 3: Comparing standard classifiers with RADIAL classifiers.

We compare with a simple one-vs-rest logistic regression classifier. Specifically, we use the LIBLINEAR library, with its default regularization (LIBLINEAR-R) and no regularization (LIBLINEAR-NR). We also compare with a kNN classifier, for $k=11$ that we found to be a good choice. We have also performed experiments with a standard SVM classifier which did not perform as well. We note that these classifiers do not use validation; the validation sets are added to the training sets.

• LIBLINEAR-R was the classifier of choice in DEEPWALK (Perozzi et al., 2014), where it was also used in combination with spectral embeddings. We see that RADIAL classifiers increase accuracy, sometimes by more than 15% relative to this previously used standard classifier [3].

• We also observe that the use of regularization is detrimental to the accuracy of LIBLINEAR, which corroborates the theoretical insight on the value of the radial projection step. However, CITESEER and CORA are exceptions, indicating that overfitting may be an issue for these smaller datasets.

---

[3](Perozzi et al., 2014) do not give details on their implementation of supervised spectral clustering and they do not use these 6 benchmarks. Based on their description, accuracy of their variant of supervised spectral clustering would have likely been lower that what we report in row-1.

• Overall the results of this section show that RADIAL classifiers perform better than standard classifiers, with the notable exception of CORA, that we discuss separately, in section 3.4.

## 3.4 RADIAL SPECTRAL CLUSTERING VS OTHER METHODS

In this section we compare the RADIAL classifier with other methods. The goal of this section is not to claim that RADIAL classifiers are superior, but to compare them directly with some widely acclaimed methods. Hence, the list is not exhaustive by any means.

| | CLASSIFIER | CITESEER | | | | | | CORA-mod | | | | | | WIKICS | | | | | |
|---|---|---|---|---|---|---|---|---|---|---|---|---|---|---|---|---|---|---|---|
| | | 1% | 2% | 4% | 8% | 16% | 32% | 1% | 2% | 4% | 8% | 16% | 32% | 1% | 2% | 4% | 8% | 16% | 32% |
| 1 | CONIC-2C | 48.1 | 60.9 | 65.3 | 68.2 | 69.9 | 71.1 | 57.9 | 63.6 | 67.9 | 75.5 | 77.1 | 79.9 | 70.5 | 74.0 | 75.0 | 76.1 | 78.4 | 79.8 |
| 2 | RANDWALK | 60.7 | 65.0 | 67.1 | 70.0 | 71.3 | 73.0 | 68.5 | 72.8 | 77.4 | 79.6 | 81.9 | 83.1 | 21.3 | 32.2 | 42.6 | 63.8 | 73.7 | 79.1 |
| 3 | GCN | 41.3 | 44.5 | 55.9 | 60.5 | 65.8 | 68.9 | 15.3 | 16.3 | 16.6 | 16.3 | 16.0 | 16.8 | 65.1 | 69.1 | 72.7 | 76.4 | 78.0 | 79.0 |
| 4 | NETMF | 39.4 | 49.8 | 53.3 | 54.5 | 57.1 | 60.8 | 61.2 | 66.1 | 69.5 | 71.6 | 74.8 | 76.8 | 70.3 | 68.5 | 68.6 | 69.5 | 73.6 | 80.7 |
| 5 | DEEP WALK | 49.3 | 47.1 | 54.1 | 63.1 | 65.8 | 67.4 | 61.9 | 69.2 | 73.7 | 77.4 | 78.5 | 79.0 | 72.1 | 75.0 | 77.0 | 78.2 | 79.2 | 79.8 |
| | CLASSIFIER | PUBMED | | | | | | ARXIV | | | | | | PRODUCTS | | | | | |
| | | 1% | 2% | 4% | 8% | 16% | 32% | 1% | 2% | 4% | 8% | 16% | 32% | 1% | 2% | 4% | 8% | 16% | 32% |
| 1 | CONIC-2C | 75.2 | 75.3 | 76.0 | 76.4 | 76.8 | 76.8 | 58.6 | 60.7 | 62.6 | 64.1 | 64.9 | 65.4 | 83.5 | 83.8 | 83.9 | 84.0 | | |
| 2 | RANDWALK | 66.8 | 73.0 | 77.0 | 79.4 | 80.8 | 81.8 | 32.3 | 41.3 | 52.2 | 60.9 | 66.0 | 69.3 | 79.1 | 83.8 | 85.3 | 86.7 | 87.9 | 89.0 |
| 3 | GCN | 69.0 | 72.8 | 75.3 | 77.3 | 79.1 | 80.3 | 56.8 | 60.1 | 62.1 | 63.6 | 64.3 | 64.9 | | | | | | |
| 4 | NETMF | 75.7 | 76.1 | 76.5 | 76.6 | 76.7 | 76.7 | | | | | | | | | | | | |
| 5 | DEEP WALK | 77.2 | 77.3 | 77.7 | 77.8 | 77.9 | 78.0 | | | | | | | | | | | | |

Figure 4: Comparing other methods with RADIAL classifiers. Computing the NETMF and DEEPWALK embeddings is extremely time-consuming for the two largest sets, thus no results are reported for them.

Specifically we consider these algorithms: DeepWalk (Perozzi et al., 2014), NetMF (Qiu et al., 2018b), a standard GCN model[4] (Kipf & Welling, 2017) and the RANDOMWALKER algorithm (harmonic label interpolation) (Grady, 2006). RANDOMWALKER is a non-parameteric algorithm and it is much faster than all other methods. On the other hand, DEEPWALK and NETMF are computationally much more expensive than RADIAL. We observe that:

• Standard GCNs are less competitive in the context of graphic data. Thus we can conjecture that GCNs make sub-optimal use of graphic features.

• Due to the approximation loss reflected in standard Cheeger inequalities, eigenvectors of the normalized Laplacian do not always yield good approximations of low-conductance clusters. According to theory developed in (Koutis & Le, 2019), in such cases, a graph modification can alleviate the problem. CORA-mod is such a modified version of the CORA graph[5]. Thus, the fact that CORA-mod gives rise to a much better spectral embedding explains the failure observed for CORA in previous sections.

• RANDOMWALKER has always a very strong performance in higher training ratios, but it can be very erratic in lower training ratios, where non-trainable graph embeddings do better. In general, RADIAL is directly competitive to DEEPWALK and NETMF, for a small fraction of runtime.

• RANDOMWALKER is based on a harmonic interpolation objective similar to that used in the Correction & Smooth (C&S) procedures in (Huang et al., 2021). Indeed, in Figure 9 in the Appendix, we observe that the accuracy enhancement added by C&S correlates strongly with the training ratio. Figure 9 also shows that an improved 'base' generally improves the final output of C&S.

---

[4]GCNs by design operate with node features; here we use one-hot vectors as suggested in (Kipf & Welling, 2017).
[5]CORA-mod is used only for CONIC-2C. DEEPWALK and NETMF effectively work with a modified graph.

## 4 HANDLING NODE FEATURES

In this section we go beyond graphic features, and include existing node features in RADIAL. To enable our RADIAL classifiers to work with features we incorporate one extra channel to the architecture of Figure 1. The features are standardized to have zero mean and constant standard deviation, and the rest of the layers are kept in place, including the Radial Projection layer. We find that this simple change raises the accuracy of our model in all cases, except the CITESEER data set. These results are reported in Figure 5.

| | CLASSIFIER | CITESEER | | | | | | CORA | | | | | | WIKICS | | | | | |
|---|---|---|---|---|---|---|---|---|---|---|---|---|---|---|---|---|---|---|---|
| | | 1% | 2% | 4% | 8% | 16% | 32% | 1% | 2% | 4% | 8% | 16% | 32% | 1% | 2% | 4% | 8% | 16% | 32% |
| 1 | CONIC-2C | 48.1 | 60.9 | 65.3 | 68.2 | 69.9 | 71.1 | 17.1 | 17.9 | 20.2 | 21.7 | 21.7 | 27.0 | 70.5 | 74.0 | 75.0 | 76.1 | 78.4 | 79.8 |
| 2 | CONIC-2C-FT | 44.2 | 51.6 | 60.2 | 63.4 | 67.5 | 69.2 | 21.3 | 26.9 | 34.5 | 43.4 | 45.0 | 52.7 | 71.8 | 73.9 | 75.3 | 77.6 | 78.0 | 80.1 |
| | CLASSIFIER | PUBMED | | | | | | ARXIV | | | | | | PRODUCTS | | | | | |
| | | 1% | 2% | 4% | 8% | 16% | 32% | 1% | 2% | 4% | 8% | 16% | 32% | 1% | 2% | 4% | 8% | 16% | 32% |
| 1 | CONIC-2C | 75.2 | 75.3 | 76.0 | 76.4 | 76.8 | 76.8 | 58.6 | 60.7 | 62.6 | 64.1 | 64.9 | 65.4 | 83.5 | 83.8 | 83.9 | 84.0 | | |
| 2 | CONIC-2C-FT | 75.6 | 76.2 | 79.2 | 80.7 | 82.4 | 85.4 | 60.0 | 61.5 | 64.2 | 65.7 | 67.2 | 68.6 | 83.5 | 84.6 | 85.4 | 85.7 | | |

Figure 5: Adding features to a RADIAL classifier

### 4.1 BOOSTING GCNS WITH LEARNED GRAPHIC FEATURES

We have found that GCNs make a better use of node features and in fact produce higher accuracy relative to that reported in Figure 5 for CONIC-2-FT. Thus GCNs remain very strong when node features are available. Given that RADIAL performs better than GCNs in the graphic setting, we thus propose to use the **learned features** of our CLASSIFIER. Specifically we use the activations of the Concatenation layer of the model learned in Figure 5, and feed them as *additional* input to the standard GCN. The outcome of this approach for the ARXIV data set is shown in Figure 6.

| | 1% | 2% | 4% | 8% | 16% |
|---|---|---|---|---|---|
| REGULAR-GCN | 63.0 | 65.3 | 67.7 | 69.5 | 71.1 |
| CONIC-BOOSTED GCN | 64.9 | 67.5 | 69.0 | 70.7 | 72.1 |

Figure 6: Enhancing node features with features learned by CONIC-2C-FT, for the ARXIV data set.

A thorough exploration of the power of these learned graphic features in the broader context of GNNs is not in the indented scope of this work, and it is left open for future work.

## 5 CONCLUSION

Research on transductive node classification tasks has recently been dominated by Graph Neural Networks and the evolution of progressively more elaborate and highly-parameterized neural architectures that yield end-to-end trainable network embeddings. This is in stark contrast to the pre-GNN period, when the collective focus was on the design of network embedding methods, with apparently little research devoted to the downstream classification task. In this work we argued that developing a better theoretical understanding of the geometric embedding opens up the fundamental research direction of designing better geometry-aware classifiers. This direction can lead to surprising substantial improvements, even for older algorithms, such as spectral clustering. In principle, this reasoning can be applied to other existing network embedding methods with the potential to –more broadly– inform the design of improved graph neural architectures.

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

## A  Datasets, Hyperparameters, Variance.

**Datasets.** We use 6 labeled datasets that have been used widely as benchmarks. These datasets have been processed to remove small connected components because spectral clustering naturally works on a connected graph[6]; self-loops are also removed. For all datasets, we treat the graphs as undirected and only consider the largest connected component. Table 1 summarizes the number of vertices, edges and classes of these largest connected components.

| Dataset | CITESEER | CORA | WIKICS | PUBMED | ARXIV | PRODUCTS |
|---|---|---|---|---|---|---|
| $|V|$ | 3,327 | 2,708 | 11,701 | 19,717 | 169,343 | 2,449,029 |
| $|\tilde{V}|$ | 2,120 | 2,485 | 11,311 | 19,717 | 169,343 | 2,385,902 |
| $|E|$ | 4,614 | 6,632 | 215,863 | 44,326 | 2,315,598 | 123,718,152 |
| $|\tilde{E}|$ | 3,679 | 5,069 | 215,554 | 44,324 | 2,315,598 | 123,612,606 |
| #classes | 6 | 7 | 10 | 3 | 40 | 47 |

Table 1: Number of vertices and edges in the original and processed graphs

**Code and Experiments.** All experiments were performed on Google Colab. The final paper will disseminate all code, including the models and the experimental setting.

**Hyperparameter Settings.** In RCS-CLASSIFY (algorithm 1), we take $k = 2 \cdot nc$, where $nc$ is the number of classes. In non-reported experiments we have found that further increasing the number of dimensions can slightly increase accuracy, but we take a theory-consistent decision to make the dimension proportional to number of clusters. The Random Walker algorithm does not have any hyperparameters or learnable parameters. For the GCN model we used the recommended settings, using the identity matrix in the input into the GCN. For NetMF we followed the recommended default settings, using an embedding of 128 dimensions. The dimension for DeepWalk is set to be $2 \cdot nc$; this deviates from the recommendation but it is added here for direct comparison with the setting for RADIAL.

**Variance.** We note that we omit information on the standard deviation of accuracy over random splits and random initializations. This is in general low ($<0.5\%$) with the exception of 1-2% training ratio for the three smaller datasets, where standard deviation can be as high as 3%.

## B  Further Discussion

• We have also considered the applications of Support Vector Machines on the radially projected spectral embedding. The results were not competitive, and thus they are not included here.

• The RANDOMWALKER method has rarely -if ever- been used as benchmark for supervised node classification on social graphs. However the success of C&S, a closely related algorithm, motivates the experimental study of the RANDOMWALKER algorithm, the strengths of which include the fact that it does not have any hyperparameters or trainable parameters. We see that in multiple cases RANDOMWALKER is a viable alternative to other models, especially considering that with a proper implementation using Laplacian solvers, RANDOMWALKER is **by far faster** than all other methods in Table **??**. The C&S process is not applicable to the output of RANDOMWALKER for certain concrete technical reasons, but a worthy research topic it to modify RANDOMWALKER in a way that would enable the application of C&S, in order to enhance its performance.

---

[6]One can use the node features to connect these smaller connected components. We avoid this practice in order to stick with the decision to only use graphic data.

## C EFFICIENT COMPUTATION

We discuss the computation of eigenvectors of the generalized eigenvalue problem $Lx = \lambda Dx$. Several works compute eigenvectors using implementations of ARPACK (Lehoucq et al., 1997) in libraries for sparse matrix algebra, in Matlab, Python, or Julia (e.g. (Huang et al., 2021)). Social networks usually have eigenvalue distributions that are 'friendly' to power methods (that are used in ARPACK) in the sense that they can converge in a reasonable amount of time. However that is not guaranteed, and in practice convergence can be much slower than possible. Fast convergence can presently be guaranteed only via using inverse-power algorithms, i.e. algorithms that use approximations of the inverse of the Laplacian, often used as preconditioners. In this work we use the preconditioned eigensolver LOBPCG (Knyazev, 2001) with the CMG preconditioner (Koutis et al., 2011). Besides its stronger theoretical properties, the particular implementation is also significantly faster in practice, even on social networks.

### C.1 INITIALIZATION AND CONVERGENCE

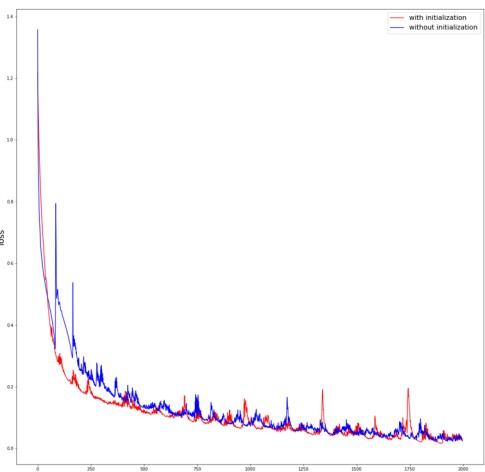

Figure 7: Convergence of the loss function on the ARXIV with and without the special initilization described in section 2.3. Initialization results in faster convergence. (Loss is in logarithmic scale for clarity.)

# D  THE ALGORITHM

---

**Algorithm 1** Radial-Conic-Spherical Classifier with $K$ channels

---

1: **Input:** $x \in \mathbb{R}^{1 \times k}$: eigenvector embedding, *type*: 'Linear' or 'Conic'
2: **Output:** $y \in \mathbb{R}^{1 \times nc}$: $nc$ is the number of classes
3:
4: **Trainable parameters:**
5: $R_i \in \mathbb{R}^{k \times nc}$, for $i = 1, \ldots K$, where $K$ is a hyperparameter
6: $W \in \mathbb{R}^{K \cdot nc \times nc}$
7: $\beta \in \mathbb{R}^{1 \times nc}$
8:
9: **function** RCS-CLASSIFY$(x)$
10:      **for** i=1 **do** to K
11:          $y = xR_i$                                                        ▷ [optional]
12:          $y_i = y/||y||_2$                                     ▷ radial projection
13:      **end for**
14:      $y = [y_1; \ldots; y_K]$                                   ▷ concatenate
15:      **if** type = 'Conic' **then**
16:          $w_{nrm} = norm(W, dim = 1)$      ▷ $w_{nrm} \in \mathbb{R}^{1 \times nc}$: vector of 2-norms of columns of $W$
17:          $y = yW - w_{nrm}.* cos(\beta)$               ▷ $.*$ denotes pointwise multiplication
18:      **else if** type = 'Linear' **then**
19:          $y = yW + \beta$
20:      **else if** type = 'Spherical' **then**
21:          $y = \beta^2 - (W - \mathbf{1}^{K \cdot nc \times 1} y)^2$            ▷ squaring is entry-wise
22:      **end if**
23:      $y = \mathbf{softmax}(y)$
24: **end function**

---

# E   MISC EXPERIMENTS

## E.1   ABLATION EXPERIMENTS

We have performed these additional ablation experiments:

- We added bias to the Reduce layer. This makes the models more general. However, in practice, it has a **negative** impact on accuracy.
- We made the Reduce layer non-trainable and orthogonal (as suggested in Lee et al. (2014a)). This has a significantly **negative** impact on accuracy.
- We have added a regularization term $||R * R^T - I||_F$ that penalizes the Reduce layer for deviating from orthogonality. This has a **negative** impact on accuracy.

We also experimented with adding more channels to the RADIAL classifiers, which seems to have a **positive impact on accuracy**. On a specific fixed training set for the ARXIV recorded accuracies are shown in Figure 8.

| ARXIV with multiple Channels | | | | | | |
|---|---|---|---|---|---|---|
| #nchannels | 1 | 2 | 3 | 5 | 6 | 6 |
| accuracy | 64.2 | 65.7 | 66.3 | 66.7 | 66.8 | 66.9 |

Figure 8: An experiment with multiple channels

## F  ALL RESULTS WITH CORRECTION AND SMOOTH

| | CLASSIFIER | CITESEER 1% | 2% | 4% | 8% | 16% | 32% | CORA 1% | 2% | 4% | 8% | 16% | 32% | WIKICS 1% | 2% | 4% | 8% | 16% | 32% |
|---|---|---|---|---|---|---|---|---|---|---|---|---|---|---|---|---|---|---|---|
| 1 | No-Radial | 56.4 | 50.3 | 51.0 | 54.3 | 57.4 | 58.4 | 16.7 | 23.9 | 25.7 | 28.6 | 26.8 | 29.0 | 29.7 | 34.3 | 35.0 | 37.7 | 35.8 | 35.9 |
| | No-Radial+CS | 58.4 | 64.5 | 69.3 | 70.4 | 72.9 | 74.2 | 69.5 | 74.2 | 75.6 | 78.3 | 81.4 | 83.2 | 52.1 | 52.2 | 54.8 | 57.3 | 60.0 | 71.5 |
| 2 | No-Reduce | 49.0 | 63.6 | 66.9 | 68.9 | 68.9 | 71.1 | 18.9 | 18.8 | 18.8 | 17.5 | 25.5 | 27.4 | 69.4 | 71.7 | 73.9 | 74.8 | 75.8 | 76.2 |
| | No-Reduce+CS | 57.6 | 64.9 | 68.7 | 70.6 | 72.2 | 74.1 | 69.3 | 73.6 | 75.2 | 77.9 | 82.7 | 84.8 | 69.2 | 73.6 | 76.9 | 77.6 | 80.0 | 81.8 |
| 3 | Conic | 45.8 | 59.6 | 64.6 | 67.6 | 69.7 | 70.7 | 16.0 | 19.0 | 19.0 | 22.7 | 24.4 | 26.7 | 69.9 | 73.2 | 75.0 | 75.4 | 78.2 | 78.7 |
| | Conic+CS | 55.9 | 64.3 | 67.7 | 70.2 | 72.1 | 74.0 | 64.9 | 72.2 | 75.1 | 80.1 | 82.7 | 84.7 | 71.1 | 74.3 | 77.8 | 78.6 | 80.5 | 81.9 |
| 4 | Spherical | 45.0 | 62.3 | 65.1 | 67.9 | 69.0 | 70.9 | 18.1 | 19.6 | 22.3 | 24.5 | 26.3 | 28.1 | 68.8 | 73.2 | 75.3 | 76.9 | 78.5 | 79.0 |
| 5 | Conic-2C | 48.1 | 60.9 | 65.3 | 68.2 | 69.9 | 71.1 | 17.1 | 17.9 | 20.2 | 21.7 | 21.7 | 27.0 | 70.5 | 74.0 | 75.0 | 76.1 | 78.4 | 79.8 |
| | Conic-2C+CS | 55.9 | 63.5 | 67.2 | 69.2 | 71.3 | 73.7 | 56.9 | 42.2 | 73.3 | 79.2 | 82.0 | 84.8 | 70.3 | 74.0 | 76.9 | 77.6 | 80.3 | 82.3 |
| 6 | Conic-2C-PT | 53.5 | 57.4 | 65.5 | 69.1 | 70.8 | 71.0 | 17.4 | 20.1 | 18.4 | 20.8 | 22.0 | 25.8 | 71.8 | 73.9 | 75.3 | 77.6 | 78.0 | 80.1 |
| 7 | LibLinear-R | 55.3 | 64.4 | 67.8 | 68.7 | 69.8 | 70.9 | 63.6 | 68.8 | 71.5 | 74.1 | 75.0 | 76.0 | 72.8 | 73.6 | 74.9 | 75.4 | 76.1 | 76.3 |
| 8 | LibLinear-NR | 54.5 | 56.7 | 65.0 | 68.6 | 70.4 | 71.1 | 64.4 | 65.1 | 71.0 | 75.7 | 78.2 | 78.3 | 67.5 | 71.0 | 75.0 | 76.2 | 76.9 | 77.0 |
| 9 | kNN | 49.6 | 62.2 | 67.3 | 68.7 | 69.8 | 70.7 | 50.9 | 66.9 | 70.7 | 74.8 | 77.3 | 78.9 | 70.3 | 72.7 | 74.6 | 76.6 | 78.5 | 79.5 |
| | kNN+CS | 57.8 | 62.9 | 65.9 | 67.8 | 69.5 | 72.1 | 71.7 | 75.2 | 76.3 | 79.9 | 82.4 | 84.4 | 73.2 | 74.6 | 76.4 | 78.7 | 80.4 | 81.4 |
| 10 | RandWalk | 60.7 | 65.0 | 67.1 | 70.0 | 71.3 | 73.0 | 68.5 | 72.8 | 77.4 | 79.6 | 81.9 | 83.1 | 21.3 | 32.2 | 42.6 | 63.8 | 73.7 | 79.1 |
| 11 | GCN | 41.3 | 44.5 | 55.9 | 60.5 | 65.8 | 68.9 | 15.3 | 16.3 | 16.6 | 16.3 | 16.0 | 16.8 | 65.1 | 69.1 | 72.7 | 76.4 | 78.0 | 79.0 |
| 12 | NetMF | 39.4 | 49.8 | 53.3 | 54.5 | 57.1 | 60.8 | 61.2 | 66.1 | 69.5 | 71.6 | 74.8 | 76.8 | 70.3 | 68.5 | 68.6 | 69.5 | 73.6 | 80.7 |
| 13 | Deep Walk | 49.3 | 47.1 | 54.1 | 63.1 | 65.8 | 67.4 | 61.9 | 69.2 | 73.7 | 77.4 | 78.5 | 79.0 | 72.1 | 75.0 | 77.0 | 78.2 | 79.2 | 79.8 |

| | CLASSIFIER | PUBMED 1% | 2% | 4% | 8% | 16% | 32% | ARXIV 1% | 2% | 4% | 8% | 16% | 32% | PRODUCTS 1% | 2% | 4% | 8% | 16% | 32% |
|---|---|---|---|---|---|---|---|---|---|---|---|---|---|---|---|---|---|---|---|
| 1 | No-Radial | 53.2 | 53.6 | 59.0 | 55.2 | 58.8 | 58.5 | 21.2 | 26.5 | 26.0 | 24.6 | 25.8 | 25.1 | 27.9 | 28.7 | 31.1 | 28.9 | | |
| | No-Radial+CS | 76.0 | 77.9 | 79.1 | 80.0 | 80.7 | 81.5 | 53.2 | 59.8 | 63.7 | 66.0 | 68.4 | 70.4 | 83.7 | 83.6 | 83.8 | 85.0 | | |
| 2 | No-Reduce | 63.5 | 64.6 | 63.4 | 64.3 | 65.2 | 64.0 | 47.6 | 48.5 | 49.0 | 49.3 | 49.3 | 49.4 | 66.6 | 66.7 | 66.6 | 66.7 | | |
| | No-Reduce+CS | 76.0 | 77.7 | 79.4 | 80.5 | 81.4 | 82.3 | 56.4 | 61.0 | 64.8 | 67.4 | 69.6 | 71.3 | 84.8 | 85.7 | 86.5 | 87.4 | | |
| 3 | Conic | 75.6 | 75.4 | 76.0 | 76.6 | 76.6 | 76.6 | 58.4 | 60.5 | 61.3 | 62.1 | 62.4 | 62.5 | 80.9 | 80.9 | 80.9 | 81.0 | | |
| | Conic+CS | 76.6 | 77.4 | 78.3 | 79.9 | 81.0 | 82.3 | 61.0 | 63.3 | 65.4 | 67.0 | 69.5 | 71.3 | 85.6 | 86.2 | 86.8 | 87.2 | | |
| 4 | Spherical | 75.0 | 75.0 | 75.5 | 76.0 | 76.1 | 76.3 | 57.8 | 60.5 | 61.9 | 63.0 | 63.7 | 63.8 | 82.1 | 82.3 | 82.4 | 82.5 | | |
| 5 | Conic-2C | 75.2 | 75.3 | 76.0 | 76.4 | 76.8 | 76.8 | 58.6 | 60.7 | 62.6 | 64.1 | 64.9 | 65.4 | 83.5 | 83.8 | 83.9 | 84.0 | | |
| | Conic-2C+CS | 76.6 | 77.4 | 78.3 | 79.9 | 81.0 | 82.3 | 61.1 | 63.3 | 65.6 | 67.6 | 69.6 | 71.3 | | | | | | |
| 6 | Conic-2C-PT | 73.0 | 75.9 | 76.1 | 76.5 | 77.4 | 77.2 | 59.5 | 61.2 | 62.9 | 64.4 | 65.2 | 65.7 | | | | | | |
| 7 | LibLinear-R | 60.3 | 60.6 | 61.7 | 64.3 | 67.1 | 68.8 | 43.8 | 45.4 | 47.3 | 48.9 | 50.2 | 51.4 | 67.1 | 69.4 | 73.4 | 75.4 | 76.5 | 77.3 |
| 8 | LibLinear-NR | 73.3 | 73.4 | 72.7 | 72.1 | 71.6 | 71.5 | 56.9 | 58.7 | 59.2 | 59.1 | 59.2 | 59.2 | 79.9 | 79.9 | 80.0 | 80.1 | 80.1 | 80.1 |
| 9 | kNN | 73.5 | 74.7 | 75.4 | 76.3 | 77.3 | 78.3 | 54.7 | 57.3 | 59.2 | 60.9 | 62.6 | 64.3 | 80.7 | 81.9 | 83.0 | 84.1 | 85.0 | 85.9 |
| | kNN+CS | 76.0 | 76.6 | 77.9 | 79.4 | 80.7 | 81.9 | 59.5 | 62.4 | 64.8 | 66.8 | 68.7 | 70.6 | | | | | | |
| 10 | RandWalk | 66.8 | 73.0 | 77.0 | 79.4 | 80.8 | 81.8 | 32.3 | 41.3 | 52.2 | 60.9 | 66.0 | 69.3 | 79.1 | 83.8 | 85.3 | 86.7 | 87.9 | 89.0 |
| 11 | GCN | 69.0 | 72.8 | 75.3 | 77.3 | 79.1 | 80.3 | 56.8 | 60.1 | 62.1 | 63.6 | 64.3 | 64.9 | | | | | | |
| 12 | NetMF | 75.7 | 76.1 | 76.5 | 76.6 | 76.7 | 76.7 | | | | | | | | | | | | |
| 13 | Deep Walk | 77.2 | 77.3 | 77.7 | 77.8 | 77.9 | 78.0 | | | | | | | | | | | | |

Figure 9: Summary of all results

