# OpenReview forum: "Neurally boosted supervised spectral clustering"
_ICLR.cc/2022/Conference — ICLR 2022 Submitted_

### Official Review · Reviewer_YsLB · 2021-10-29

**Correctness:** 4
**Technical Novelty And Significance:** 1
**Empirical Novelty And Significance:** Not applicable
**Recommendation:** 3
**Confidence:** 4

**Main Review:**

Strengths:
1. This paper is very clear and easy to follow and understand.
2. It conducted solid experiments to show the performance differences for each method.

Weakness:
It is very difficult to position and evaluates this paper. Though it concretely introduces lots of methods, the paper itself does not propose any new methods or insights, such that it does not qualify as a "new-method" paper. As for the "review & benchmark" paper, this paper lacks a more detailed exploration of this topic and the experimental validation is rather poor compared with review papers. Still, I will treat this paper as a review paper and below are some of my unuseful comments.

1. The mathematical background could be more concrete. Instead of briefly introducing the cut and conductance, the author would also show the equation of graph Ncut, which reveals the power and need of conducting spectral clustering for node embedding.

2. While introducing related methods, it would be better to use mathematical equations to showcase their insights and improvement compared with previous methods.

3. The "orthogonal projection", "radial projection" and "geometric partitioning" can be unified in the classifier section. All these projections could be considered as the preprocessing for classifiers. The author could consider introducing them in the neural classifier part altogether.

4. Though section 2.1.3 talks about clustering from the geometric perspective. But it does feel disconnected in this paper. I don't see the benefit here.

5. About the efficient computing of the eigendecomposition, the author could consider eigengame (ICLR 2020).

6. For a more comprehensive experiments. The author could consider compare spectral clustering embedding with other embedding. Like node2vec, and more graph neural network based methods like GAT, GraphSage et al.





**Summary Of The Paper:**

This paper reviewed the application of spectral clustering embedding on node-level classification problems. It first revisited the theoretical foundation of spectral clustering, related to the graph NCut problems. Then it introduced neural network-based classifiers which utilize the embedding for classification tasks. Their benchmark results validated that the neural network framework showed certain advantages in accuracy. And the "correction & smooth" introduced by Huang et al (ICLR 2021) would also improve the performance in a relative big margin.

**Summary Of The Review:**

This paper is very well written. But it lacks originality and also did not derive new insights.

---

> ### Author Response · Authors · 2021-11-09
> **new method paper (?)**
>
> The reviewer raised a concern that the paper cannot be considered a new method paper. But the entire Algorithm 1 is a *new* method that has not been proposed or published before. For example, the Conic and Spherical classification surfaces, the multiple channels, and the specially designed reduce layers, along with their initializations (section 2.3) are all new. It is true that our algorithm derives from original insights of the *unsupervised* algorithm of Lee et al, but almost none of the components of Algorithm 1 have appeared before. Also, multiple rows in Table 1 are not just benchmarking, but showcase the extremely improved performance of a fundamental algorithm, that was never observed before.  We also believe that the work has elements of originality -- can the reviewer find other examples of using conic and spherical classification surfaces in an effective way, or can they find references to them in any other work?
>
> Thus, with all due respect, we disagree with the reviewer that this is not a new-methods paper.

---

### Official Review · Reviewer_GXsL · 2021-10-29

**Correctness:** 3
**Technical Novelty And Significance:** 2
**Empirical Novelty And Significance:** 2
**Recommendation:** 5
**Confidence:** 4

**Main Review:**

==STRENGHTS==

BACKGROUND
I have found the theory well written and self-contained. I think a non-expert could find most of the information in the paper, and I appreciate this aspect.

INSIGHTS
The geometrical insights are didactical and well communicated. The conclusion given by the experiments looks interesting and valuable for future practitioners, while I think a synthesis would be beneficial for the reader.

==WEAKNESSES==

NOVELTY AND CONTRIBUTION
This work does not present any particular methodological novelty, and I see this paper more as an experimental one. I think such kinds of papers are valuable and deserve attention, but in this specific case, I think the contribution provided by the analysis is limited in the following aspects:
a) there are no specific comments for performances on different datasets; for example, why on CORA does the "Conic" perform poorly without "+CS"? Why on "arXiv" outperforms all other methods? Which are the peculiarities of the different datasets? I see that such analysis may be complicated given the dimensionality of the graphs, but I think that even some insights given by further investigation would be important
b) the experiments lack variety in the applicative domain; for example, spectral clustering has obtained significant attention in the 3D field. It would be interesting to apply the techniques to at least one geometrical dataset that is more prone to have symmetries. These latter are the main limitations of spectral methods that do not consider additional features and should be somehow discussed in the paper to provide a complete overview.
c) I think that considering only the quantitative performances tells only part of the story, and a deeper analysis should be provided (e.g., error localization, timings)

PRESENTATION
Apart from the background, I find the rest of the paper a bit verbose, and that could be improved in its communication. For example, visualization of spectral embeddings and a pipeline scheme would help a lot in diving into the paper. Also, It is not comfortable to move back and forth to check the results of the final table: I suggest splitting it in two and locate near the discussion text. I think the visual aspect of the paper could be significantly improved and serve as support to the analysis.

**Summary Of The Paper:**

The paper proposes a study about spectral clustering techniques. Given the standard spectral clustering approach, some variations are investigated, also using recent advancements in the field. The experiments show that such a simple approach can produce state-of-the-art results on social graphs.

**Summary Of The Review:**

While I like the general idea and I appreciated the effort in explaining the underlying theory, I think there is no significant methodological novelty and also a significant lack in the experimental and analysis settings. I do not consider it ready for publication in the present state.

---

### Official Review · Reviewer_7yDC · 2021-11-02

**Correctness:** 3
**Technical Novelty And Significance:** 2
**Empirical Novelty And Significance:** 2
**Recommendation:** 3
**Confidence:** 4

**Main Review:**

Strengths: The paper is mostly well written, and the main thrust of the paper is an important point for researchers in this area: simple, but well structured and theoretically motivated graph models can perform surprisingly well against larger, more complex models, and can even outperform them. Also, the empirical results are presented honestly, which is greatly appreciated.

Weaknesses: The main weaknesses of the paper are: (1) a lack of direction and (2) an insufficient distinction from previous works.

Is the point of the paper to alert the community to the previous work of Lee et al. 2014, or is it to emphasise that such models have impressive empirical performance compared with other methods? If it is the former, then Section 2 can be improved by clearing up notation and errors as well as more firmly delineating where this paper diverges from existing work, which would help with point (2) as well. Is Section 2.1.3 the insight of the authors or does it also summarise previous works?

If it is the latter, then I do not find the argument of the paper either novel or convincing. The results demonstrate that the addition of correction and smooth, the essential point of Huang et al. 2021, consistently has the largest impact on the accuracy. A counter argument to the paper’s results is that the node classification tasks in Citeseer, Cora, and Pubmed are in a way too simple, being well described by local smoothing of the labels. If that is the case how does this differ from the point of Huang et al.? Recent GNN research has focused on topics such as incorporating additional node and edge information, which the proposed method could attempt to incorporate, but is not included in the empirical section as the authors choose to focus solely on the connectivity information. How do the proposed models compare when such information is considered? How can these data be incorporated into the models in a principled manner? The other direction for more challenging graph problems is in inter-domain settings, where there are multiple graphs (such as the molecule classification benchmarks) or in finding correspondences between nodes in situations where the graph has been modified in some way (by pruning or modifying the ambient embedding). Unfortunately, the proposed methods would fail in these situations. However, this would be a very interesting direction for the authors to pursue in the future.

What exactly is ‘neural’ about the additional classifiers? Such a description seems to detract from the main claim, and does not seem justified. Is it simply that there are two steps: the spectral embedding and the classification? Is it the inclusion of the K independent channels?

Notation and definitions:

a) the definition of vol(.) should be consistent in its indexing.

b) my preference would be to reserve degree of a vertex for the unweighted count of edges, and call what is in the paper the weighted degree.

c) I am familiar with the definition of conductance, but have not seen the notation of cap(.) before, a brief definition would likely improve clarity.

d) I understand what the authors are attempting to communicate about a partition of the vertex set, but as written the statement is incorrect: the S_i must be disjoint as well.

e) Final sentence of Conic in Section 2.2, the second ‘Conic’ should be ‘Linear’.

Empirical Results:

a) Why are there empty entries for the proposed models in the Products results? Do the methods fail at this scale or are there other problems?

b) Inclusion of the results of GCN, NetMF, and DeepWalk in the Arkiv and Products tasks would aid the paper as these are much larger and could be treated as a more challenging comparison. Is there a reason these are not included?

**Summary Of The Paper:**

The paper argues for refocusing the efforts of the graph neural network community onto previous work on spectral clustering. The authors propose the addition of simple, but structured, classification methods to an existing unsupervised spectral clustering algorithm. A battery of empirical results on standard graph benchmarks demonstrate the effectiveness of such simple models and compare favourably to existing, more complex models.

**Summary Of The Review:**

I appreciate the main point of the paper, that simple models can perform surprisingly well on standard graph benchmarks, and I find such methods interesting. However, in its current form, the paper insufficiently distinguishes itself either from existing theoretical works such as the cited Lee et al., or the essential empirical point of Huang et al., and so I recommend rejection.

---

### Official Review · Reviewer_os5L · 2021-11-03

**Correctness:** 4
**Technical Novelty And Significance:** 3
**Empirical Novelty And Significance:** 3
**Recommendation:** 5
**Confidence:** 3

**Details Of Ethics Concerns:**

This paper does not introduce any new ethics concerns.

**Main Review:**

**Summary**. This paper investigates the use of spectral clustering-based approaches for node classification in graph data. It extends analysis of spectral clustering in Lee et al (2014) including the key steps of spectral embedding, orthogonal projection, radial projection, geometric partitioning.  Overall this paper provides a clear and thoughtful presentation the spectral clustering-based methods. The clarity and care the authors put into understanding each component of the algorithms is a key merit of the paper.

**Merits**. This paper does an excellent job in its thorough investigation of these spectral clustering-based approaches. Not only does this provide an array of competing methods for future work, but also provides for a better understand of the impact of various components of related models. I think this kind of careful analysis is needed, especially as the number of such methods for node classification / related problems increases. Suggesting such a critical look at past work and introduction of baselines from decades prior (e.g. RandomWalker (Kemp, 2006)), I think is fantastic an important contribution.

**Concerns**. Despite these merits, I have following concerns about the paper.
* While there is a careful analysis of the different design decisions/performance tradeoffs, I feel that there is only a limited understanding about what are the properties of the datasets that lead to these decisions/performance differences. I.e., what is about the Arxiv data that makes Conic more effective? Is it just the dataset size? Other properties? Why is Knn competitive on some dataset, but then much worse that sota on others (e.g. cora 1%)?
* While these spectral clustering methods are simpler / more scalable than some competing GCN-based approaches. I feel the (train or test) efficiency / accuracy trad-offs are not sufficiently explored? I.e. when would be scenarios/considerations that someone would chose one of these approaches?
* I think the paper would be improved with some discussion of tradeoffs empirically from a wider variety of GCN approaches.

**Minor Notes / Typos**:

* Top Table of Table 1: RandomWalker Cora 4% seems to be better than other methods, shouldn't it be bolded? Same for knn in WikiCS 1%?
* Intro: "spectacular comeback" is perhaps a bit informal/colloquial?
* 2.1: "Along it" phrasing seemed confusing to me.
* 2.1.2: There seem to be four components rather than "three" as noted?
* Top Table of Table 1: 16 % and 32% seems larger font than others
* Table 1 has a lot of results. and located at the end of the paper as it is, is somewhat difficult to navigate, especially as reading through the experiments section. Perhaps there is a way to include it at least before the end of the text of the section?
* Several of the citations in the text are missing the year.





**Summary Of The Paper:**

This paper presents a comprehensive analysis of spectral clustering based approaches for node classification in graph-based data. The paper thoroughly analyzes design decisions of using spectral clustering in this supervised setting. The empirical results indicate that even compared to GCN / deeper approaches, the less parametric spectral clustering-based approach can be competitive or better at classification tasks.

**Summary Of The Review:**

This is a thoughtful investigation spectral clustering in classification problems for graph-based data. However, there are several concerns regarding missing forms of analysis such as deeper investigations into why certain methods work well on certain datasets.

---

### Official Review · Reviewer_L1xu · 2021-11-07

**Correctness:** 4
**Technical Novelty And Significance:** 4
**Empirical Novelty And Significance:** 4
**Recommendation:** 8
**Confidence:** 4

**Main Review:**

The transfer of geometric intuition from Lee at al. leads to an improved pipeline for supervised spectral-based classification, and is a nice insight. It could spark further research in this direction, by building on strong theoretical foundations.

It is quite surprising to see that such a supervised spectral clustering pipeline attains higher accuracy than a number of sota methods which were thought to be miles away better in performance, which also has lead to spectral methods being dropped from the comparison baselines in most works. This work demonstrates that with the right pre-processing steps in place (like the radial projection onto the unit sphere), spectral methods could be brought back to the comparison board.

The experimental results are convincing, though they could be made more complete, but it is enlightening to observe such performance of from a spectral based methods.

It would be good to see a comparison of performance for synthetically generated data, for example by considering standard stochastic block models.

**Summary Of The Paper:**

This paper puts forth a methodology for supervised spectral clustering, by leveraging existing theoretical foundations of spectral clustering, in particular drawing insights from seminal works such as Lee at al on multi-way Cheeger inequalities.


**Summary Of The Review:**

An interesting paper which could open further avenues of research at the intersection of spectral methods and neural-based approaches.

---

### Decision · Program_Chairs · 2022-01-20

**Decision:**

Reject

**Comment:**

There was some discussion on this paper, both with the authors and between reviewers. On the one hand, there is a general agreement that the empirical results suggesting that spectral clustering-based method can be competitive with SOTA methods on node classification benchmark is an interesting result. One the other hand, reviewers did not find a significantly novel contribution in the methodology proposed, and found that the empirical evaluation lacks depth and details to be really informative (eg, to understand why some methods work or not on some benchmarks). There is therefore a consensus that the paper is not ready for ICLR in its current form, but we hope that the reviews and discussion will help the authors prepare a revised version in the future.